# Batteryless, Miniaturized Implantable Glucose Sensor Using a Fluorescent Hydrogel

**DOI:** 10.3390/s21248464

**Published:** 2021-12-18

**Authors:** Hyeonkeon Lee, Jongheon Lee, Honghyeon Park, Mi Song Nam, Yun Jung Heo, Sanghoek Kim

**Affiliations:** 1Department of Electronics and Information Convergence Engineering (BK21 Four), Kyung Hee University, Yong-in 17104, Korea; hkeon2@khu.ac.kr (H.L.); Jonghun@khu.ac.kr (J.L.); 2013103922@khu.ac.kr (H.P.); 2Institute for Wearable Convergence Electronics, Kyung Hee University, Yong-in 17104, Korea; 3Department of Mechanical Engineering and Integrated Education Institute for Frontier Science & Technology (BK21 Four), Kyung Hee University, Yong-in 17104, Korea; misong333@khu.ac.kr

**Keywords:** wireless, glucose, implantable, fluorescent, WPT, batteryless

## Abstract

We propose a biomedical sensor system for continuous monitoring of glucose concentration. Despite recent advances in implantable biomedical devices, mm sized devices have yet to be developed due to the power limitation of the device in a tissue. We here present a mm sized wireless system with backscattered frequency-modulation communication that enables a low-power operation to read the glucose level from a fluorescent hydrogel sensor. The configuration of the reader structure is optimized for an efficient wireless power transfer and data communication, miniaturizing the entire implantable device to 3 × 6 mm ${}^{2}$ size. The operation distance between the reader and the implantable device reaches 2 mm with a transmission power of 33 dBm. We demonstrate that the frequency of backscattered signals changes according to the light intensity of the fluorescent glucose sensor. We envision that the present wireless interface can be applied to other fluorescence-based biosensors to make them highly comfortable, biocompatible, and stable within a body.

## 1. Introduction

Diabetes has become an epidemic affecting over 400 million people worldwide [1]. To care for diabetes, understanding blood glucose concentration is essential. The conventional blood-glucose monitoring is self-monitoring of blood glucose (SMBG), accompanying painful finger-pricking blood sampling. A newer method to monitor blood glucose is continuous glucose monitoring (CGM) that provides dynamic, continuous blood-glucose information and gives a user an alarm when abnormal blood glucose occurs. Wearable devices, such as lens-type [2,3] and bandage-type [4], and implantable sensors using glucose oxidase reactions [5,6] and glucose-sensitive permittivity change [7] were proposed for zero finger-pricking CGM. Despite their promising results, the previous sensors are not sufficient for CGM. Sweat/tear glucose sensors suffer low accuracy and low correlation with blood glucose [2,3,8]. Glucose oxidase is not suitable for long-term use within a body since the enzyme depletes with time. The permittivity of a tissue may change due to reasons other than the glucose level, exacerbating the selectivity of the sensor.

In contrast, a fluorescent glucose sensor based on boronic acids [9,10] is promising for long-term, accurate continuous glucose monitoring. Placed under the dermis, whose tissue fluid shows a higher correlation to blood than that of tears [2,3] and sweat [4], the fluorescent glucose sensor reversibly responds to glucose concentration for longer than 3 months [9,10,11], which far exceeds the lifespan of the implantable sensors based on glucose oxidase (∼14 days).

Although transdermal detection of fluorescent signals is minimally invasive [9], the skin opacity, which reduces the fluorescence intensity through the skin, and the ambient light can perturb the accurate measurement of the fluorescence intensity of glucose sensors. To overcome these issues, fully-implantable sensor systems were proposed [12,13]. For example, Tokuda et al. [13] proposed to assemble a CMOS image sensor with the fluorescent glucose sensor and then implant them under the skin. The image sensor directly read the fluorescence intensity without attenuation through skin and interference with the ambient light. However, they could not achieve wireless communication. The recent industrial effort succeeded to develop a fully implantable, wireless glucose sensor, named Eversense (3.5 × 18.3 mm), by Senseonics [11,14]. Eversense regularly saves digitized data of the fluorescent light intensity in a memory and sends the data back to a reader using Near-Field Communication (NFC). For this purpose, it includes several auxiliary components such as an Analog-to-Digital Converter (ADC), a Digital-to-Analog Converter (DAC), a nonvolatile memory, a large power-buffer capacitor, and a control state machine. Consequently, the whole implantable device size is as large as 18.3 mm in length.

On the other hand, the miniaturization of an implantable sensor is crucial to be minimally invasive for the insertion and to minimize potential discomfort as well as the immunity reaction against the sensor [15,16]. This work proposes a millimeter-scale (3 × 6 mm${}^{2}$), wirelessly powered glucose sensor device. The size could be miniaturized by removing the power-hungry components and adopting the analog backscattered modulation scheme. The reader antenna structure is configured to optimize the wireless power transfer and data communication with the coil structure printed on the sensor. The off-the-shelf photodiode component converts the light intensity into a frequency-modulated signal, which can be delivered to the external reader by backscattered modulation while the sensor is being powered. The transmit power from the reader is regulated to be under 33 dBm in order to obey the safety regulation about the Specific Absorption Rate (SAR). We then demonstrate the in vitro test to show the feasibility of CGM.

## 2. System Overview

Figure 1 shows the overall system diagram, including communication and sensing blocks. It consists of an external readout device (the left side of Figure 1) and the implantable sensor (the right side of Figure 1). The implantable device is covered with a fluorescent hydrogel and placed near the capillary under the dermis.

### 2.1. Glucose-Responsive Fluorescent Monomer

Although this work is mainly focused on electronics, the property of the fluorescent monomer is briefly provided in this section to explain the complete system. The glucose-responsive fluorescent monomer (GF-monomer) is made of diboronic acid and an anthracene [9,17]. A reactant solution of the dissolved AAm-PEG-COOH and GF-dye is evaporated and dried to obtain GF-monomer [18]. The diboronic acid can recognize and bind with the glucose molecule, while the anthracene provides the nature of fluorescence. In the absence of the glucose molecule, the unshared electron pair of the nitrogen atoms quenches the fluorescence of the anthracene, by a phenomenon known as Photo-induced Electron Transfer (PET). On the other hand, when the diboronic acid binds with the glucose molecule, the boron atom shares the electron with the nitrogen, the PET is inhibited, and the fluorescence of anthracene becomes higher. High selectivity of the diboronic acid for the glucose compared to other sugars assures the fluorescence response reflects the glucose level accurately [9]. Particularly, the diboronic acid is promising to be used as an implantable glucose detector since the reaction with the glucose molecule is reversible and does not require any reagent or enzyme. Moreover, by polymerizing it with a highly biocompatible polyacrylamide hydrogel, it can serve as the glucose detector for the long term (more than 140 days) as implanted [10].

To design the implantable frequency-modulation module, we decided the wavelength for the excitation and the emission based on the experiment. As physiological glucose concentration was varied from 0 to 500 mg/dL, we obtained excitation profiles (Figure 2a) and emission profiles (Figure 2b). The excitation profiles showed the strongest responses at a wavelength of 400–425 nm. The emission profiles showed peaks at a wavelength of 450–500 nm. When we re-plot the fluorescence intensity observed at 490-nm wavelength for varying glucose concentration, the glucose response measured by fluorescence spectroscopy is shown in Figure 2c. Thus, we can obtain high glucose sensitivity by measuring fluorescence intensities at a wavelength longer than 450 nm with the excitation wavelength of 400–425 nm. Based on the results, we chose the LED (SM0603UV-405), which produces ultraviolet (UV) light with a peak wavelength of 410 nm as an excitation light source. We also employed two optical filters to prevent direct coupling from the excitation source of LED to the photodiode bypassing the fluorescent monomer (Figure 1). A bandpass filter (ET405/40x) with a bandwidth of wavelength from 390 to 425 nm was placed in front of the LED to pass the optimized excitation light to hydrogel glucose sensors. A long-pass filter (450 nm OD 2 ultra-thin long-pass filter) with the cut-off wavelength at 450 nm was applied in front of a photodiode (TSL-238T) to have high glucose sensitivity. Note that the employment of the long-pass filter also helps to prevent the ambient light from being noise to the system. In addition, our system can be implanted to subcutaneous tissue under ∼1 mm from skin surface, which would further block the ambient light due to the opacity of skin [19].

### 2.2. Overview of Electronics

To measure the fluorescence intensity of hydrogel glucose sensors, we developed the implantable sensor including a loop antenna, a rectifier using Schottky diodes (SMS7630), a 3.3-V regulator (TPS732), and a transistor (MCH3474) to shift the load impedance of antenna for backscattered communication, in addition to the aforementioned LED (SM0603UV-405) and the light detector (TSL-238T) (Figure 1). On the top side, the LED and the light detector are placed as shown in Figure 3a. The loop antenna is printed on the bottom side of the PCB where all the other components were mounted. The total size of the implantable sensor is 3-by-6 mm${}^{2}$.

As the transmitter is excited with a carrier frequency, the voltage is induced across the loop antenna that is electromagnetically coupled to the transmitter. The induced voltage across the loop antenna is converted to DC voltage through the power management components of the rectifier, the regulator, and the storage capacitor. The regulator is employed to prevent over-voltage when excessive RF power flowed in and to stabilize the supply voltage V${}_{dd}$. The storage capacitor endures and mitigates the slump of power during backscattering by storing a large amount of charges within it.

When a stable supply voltage is provided, the light detector, which consists of a photodiode, a voltage buffer, and a pulse generator, transduces the fluorescence intensity of hydrogel glucose sensors to a frequency-modulated pulse signal. Accordingly, the pulse frequency changes depending on the glucose concentration. When the pulse turns on the transistor connected across the loop antenna, the load impedance of the loop antenna is shorted. This perturbs the impedance matching between the receiver antenna and the load; thus, the amplitude of the backscattered signal changes. As the pulse goes up and down with the frequency that encodes the glucose information, the amplitude of the backscattered signal also changes with the same frequency. This indicates that the reflected signal is frequency modulated with the pulse frequency corresponding to the glucose concentration.

Among many ways to restore the frequency by which the reflected signal is modulated, this work used a directional coupler and a spectrum analyzer. The directional coupler guides the reflected waves to the spectrum analyzer. The spectrum analyzer exhibits the frequency components of the reflected signal that includes the carrier frequency as well as the modulation frequency. The offset of the modulation frequency increases with glucose concentration, from which one can deduce the glucose level from the outside.

## 3. Wireless Link

The size of an implantable device is critical since device size affects biocompatibility, user comfort, and device failure in a body. It is essential to maximize the coupling between the transmitter (i.e., the readout device) and the receiver coil (i.e., the coil at the implanted sensor) in order to provide sufficient power to operate the circuits and enable the backscattered communication despite the small size of a device. To satisfy the requirements, we design the structure of the energy coupler to configure a loop structure. We explain the transmitter antenna configuration in Section 3.1. To demonstrate the feasibility of the wireless link, we conducted the simulation in Section 3.2.

### 3.1. Optimization of the Transmitter Antenna Configuration

The sensor coil dimension was determined as 3-by-6 mm${}^{2}$ from the available off-the-shelf components in Figure 3a, with its normal direction facing the air–tissue interface (Figure 3b). To maximize wireless power delivery through such a tiny coil, we optimized the operating frequency and the transmit structure. Since the operation frequency of a few GHz range is known to yield the highest power transfer efficiency for a millimeter-scale device [20], we operated the device at 1.5 GHz. The transmit structure could be determined from the current distribution that induces the maximum voltage at the receiver coil for a given power loss of the tissue.

In general, the fields generated by a current source J can be expressed as the convolution between the Green’s functions and the current J: (1)$$\begin{array}{cc} E_{1}\left(r\right) & =i\omega \mu_{0}\int {\bar{G}}_{E}\left(r-r^{\prime }\right)J\left(r^{\prime }\right)dr^{\prime } \end{array}$$(2)$$\begin{array}{cc} H_{1}\left(r\right) & =\int {\bar{G}}_{E}\left(r-r^{\prime }\right)J\left(r^{\prime }\right)dr^{\prime } \end{array}$$

For a planar medium, it is preferred to restrict the current to a planar source as in Figure 3b and compute the fields in the spectral domain to make Green’s operators mathematically tractable. Taking the 2-D Fourier transform in the transverse directions (x,y), the fields are the products between the current J and the Green’s functions:(3)$$\begin{array}{cc} E_{1}\left(k_{s},z\right) & =i\omega \mu_{0}{\bar{G}}_{E}\left(k_{s},z\right)J\left(k_{s}\right) \end{array}$$(4)$$\begin{array}{cc} H_{1}\left(k_{s},z\right) & ={\bar{G}}_{H}\left(k_{s},z\right)J\left(k_{s}\right), \end{array}$$
where $k_{s}=\left(k_{x},k_{y}\right)$ refers the spatial frequency in *k*-space.

A *coupling factor* γ is defined as the ratio of the open-circuited voltage to the power loss in the tissue, $\gamma :=|V_{oc}{|}^{2}/P_{loss}$. Since the received power is proportional to the square of the open-circuited voltage, the coupling factor is directly related to the power transfer efficiency [15,21]. When a small receiver coil has the area of $A_{r}$ and the normal vector $\hat{n}$, it can be shown [15] that the coupling factor is maximized when the current distributes as
(5)Jopt(ks)=∫ImϵG¯E*(ks,z)G¯E(ks,z)−1G¯H*(zf)n^.

The current distribution in the spatial domain can be obtained by taking the inverse Fourier transform of (Equation 5); $J_{opt}=F^{-1}J_{opt}$.

For the small loop at $z_{f}=$2 (mm) heading toward the air–tissue interface at $z_{1}=$1 (mm), which is comparable to the depth of commercial products, the optimal current is calculated as Figure 3c. It resembles the current of the coil and demonstrates that a loop structure that exploits the inductive coupling yields the highest efficiency [15,22]. It can be easily realized with the coil structure with a radius of 1 cm and the power can be fed from the center of the coil, by the connection with a coaxial adaptor, as shown in Figure 3d.

### 3.2. Wireless Analysis in Simulation

The connectivity between the transmit coil in Figure 3d and the receive coil in Figure 3a is examined in a commercial electromagnetic simulator, Ansys HFSS [23]. The characteristics of the link are exported to a two-port network in an s2p file format and combined with other circuitries, such as matching networks, a transistor at the receiver for signal modulation, and a directional coupler in Agilent ADS [24] for system simulation, as in Figure 1. Two-port S-parameter simulation is set to have the port impedance of 50 Ω at the transmitter side to represent the characteristic impedance of the transmission line. The spice model of a diode (SMS7630) is used at the receiver side for the Large-Signal S-parameter (LSSP) simulation of the rectifier input impedance. The matching network at the transmitter side is configured to have a 50-Ω matching at 1.5 GHz in the absence of the receiver. Similarly, the matching network at the receiver is implemented with a shunt capacitor 0.2 pF to cancel out the coil inductance at 1.5 GHz.

Including the matching network, the power gain, defined as the ratio of the received power to the transmitted power, is plotted in Figure 4a, when the modulation transistor is off. Figure 4b shows the $S_{11}$ in the Smith chart from 1 to 2 GHz when the modulation transistor is off (solid line) and on (dashed line). On each line, the $S_{11}$ values at 1.5 GHz are denoted with markers. The difference in a radial distance between two points indicates the reflected power should be different between the on and off status of the modulation transistor. As the transistor repeatedly turns on and off with time, the reflected RF signal level appearing after the directional coupler changes, as in Figure 4c (simulation results). When the modulation transistor is on for a small fraction of time, the reflected signal level slightly increases as more power is reflected back, perturbing the power delivery. To minimize the perturbation, the duty cycle of the backscatter modulation is limited to 10%, 5-μs pulse width out of 50-μs period.

## 4. System Validation

We conducted in vitro glucose-response tests. To have biocompatibility and protection layers for electronics, we encapsulated the glucose sensor system with polydimethylsiloxane (PDMS). As shown in Figure 5, the glucose sensor system was immersed in glucose solution. The height of the receiver could be adjusted to mimic the effect of depth variation. The transmitter coil in Figure 3d was placed underneath a 1-mm thick petri dish. Then, to minimize the noise from ambient light, the measurement was carried out in a dark chamber.

### 4.1. Test of Electronics and Communication

We tested the operation of electronic parts of the glucose sensor system, including the modules of the wireless power receiver, the photodiode, the pulse generation, and the backscattered communication. As shown in Figure 5, at the ceil of the dark chamber, a 1-cm diameter hole was made to provide an external, controllable light source. The light source mimics a fluorescent light scattered by the fluorescent monomer sensor. The LED and the glucose sensing hydrogel of the device were not employed for this experiment to focus on the test of electronic and communication operations. When the light intensity changes from 18 to 74 lumens, backscattered signals at the spectrum analyzer were measured as in Figure 6. The light intensity in lumens was measured in a dark room with a smartphone placed 1-m apart from the light source. The *modulation frequency*, denoted as *B* in Figure 6, increases with the light intensity. Since the distance between the transmitter and the receiver is unchanged in this experiment, the *relative amplitude* of the modulated signal to the carrier frequency, denoted as *A* in Figure 6, is not affected.

We then investigated the effect of the transmission power and the distance of the system operation at a fixed glucose concentration. The LED of the device was restored and used instead of the external light source. As the distance increased, the relative amplitude was measured for the device in the air (Figure 7a) and the glucose solution (Figure 7b). For both of them, the relative amplitude decays almost exponentially with the distance (approximately 8 dB per 1 mm). The loss within the glucose solution deteriorates the relative amplitude by another 3 dB. As the distance further increases, the power supply from the transmitter could not fulfill the power demand from the LED and the system ceases to operate. As the transmit power increases from 32 to 36 dBm, the maximum distance can enhance from 1.6 to 2.8 mm.

### 4.2. Safety Test

In practice, since an electronic device shall radiate on human skin for continuous glucose monitoring applications, the transmit power is typically limited under a few Watts due to the safety regulation [22,25,26]. We examine the allowable transmit power from the reader under the safety regulation in this subsection. To limit the RF energy exposure on the human body, the Specific Absorption Rate (SAR) is often observed in applications such as MRI and biomedical devices. The SAR is the rate at which electromagnetic energy is absorbed per unit mass by a human body. For the exposure on a partial body, the limit for the SAR is calculated as described in IEC 60601-2-33:(6)$$\begin{array}{cc} \mathrm{Partial}\;\mathrm{body}\;\mathrm{SAR} & \leq 10-\frac{8m_{exposed}}{m_{whole}}\left(W/kg\right), \end{array}$$
where $m_{exposed}$ refers to the mass of exposed body and $m_{whole}$ refers to the mass of whole body. Moreover, the regulation on the SAR for a short duration indicates that the SAR over any 10 s period shall not exceed two times this value. In the case of this study, where $m_{exposed}$ is much smaller than $m_{whole}$, 20 W/kg is the limit in the consideration of the short duration SAR.

Figure 8 shows the experimental setup and the results for the SAR measurement. The setup in Figure 8a is the same configuration as Figure 1. For the sake of mimicking a tissue, a 15-kg piece of pork was employed. The receiver was placed about 0.3 mm under the surface of the skin at the implant location of Figure 8a. The experiment measured the temperature by an infrared thermometer (CRIS-10), which captured before and after the 33 dBm RF powering as shown in Figure 8b. Because 10 s of an operation period hardly made a difference (ΔT<0.1K) in measured temperature, the period was lengthened to be 20 s. Through repetitive (n=37) measurements with the 20-second period, the average SAR was measured to be 19.6 W/kg by
(7)$$\begin{array}{cc} \mathrm{SAR} & =\frac{c\Delta T}{\Delta t}, \end{array}$$
where *c* refers to the heat capacity for epidermis of pig (3530 J/kg/K), ΔT for the temperature increase in K, and Δt for the exposure time in seconds. The result of measurement is consistent with the electromagnetic simulation results obtained from the HFSS shown in Figure 8c.

It is worthy to note that the experiment for the SAR and the temperature increase was conducted assuming a constant heat capacity without a cooling effect by the blood circulation. When one considers that the heat capacity of a live tissue increases by the blood circulation and the measurement period typically takes less than 10 s, the actual temperature increase should be less than 0.05 K per each operation with 33-dBm output power.

### 4.3. The Whole System Test

The device is to be surrounded by interstitial fluid (ISF), of which the glucose concentration is almost 90% of the venous glucose concentration [27]. Therefore, we verified the in vitro glucose responsiveness of the system in a physiological blood-glucose concentration range from 0 to 500 mg/dL. The sensor device coated with the PDMS under the florescent hydrogel was placed in a petri dish. We applied 3 mL of glucose solution in the petri dish and waited for 10 min to allow the glucose molecules to penetrate into the hydrogel and react with the fluorescent monomer. They were placed within the dark chamber in Figure 5 to remove the noise from ambient light.

The sensor device was wirelessly powered from the reader placed under the petri dish and LED light excited the fluorescent hydrogel, as shown in Figure 9a. The frequency offset of the backscattered signal was measured by the spectrum analyzer for each glucose concentration. Figure 9b shows the frequency offset signals depending on glucose concentrations, which has the same trend with Figure 2c. The overall measurements were repeated thirty times (n=30) and the spread of the measurement results were indicated by the error bars in the plot. We evaluated sensor accuracy based on the Mean Absolute Relative Difference (MARD) and the Clarke error grid analysis as in Figure 9c,d, where MARD = 8.8% and all of the points placed in the A and B zones. Typically, a CGM satisfying that the MARD is less than 10% and its sensor data are confined in the A and B zones of the Clark error grid analysis is considered as clinically acceptable. Therefore, we demonstrated a millimeter-sized implantable glucose sensor which can inform the glucose level to the external reader through a frequency-modulated backscattered signal with a clinically acceptable accuracy.

## 5. Conclusions

To minimize discomfort and an immunity reaction by implanting a fluorescence-based continuous glucose monitoring system in a body, it is crucial to minimize the size of the device. In this work, we minimized the implantable sensor system with fluorescent hydrogels to 3 ×6 mm by removing all the auxiliary components such as a battery, an ADC, and a data storage component. The power was supplied wirelessly from an external reader and physiological data read by the sensor could be directly sent to the reader using analog backscattered communication. The present mm sized system showed low transmit power in compliance with safety regulations and long distance power transmission up to 2 mm.We succeeded transferring the fluorescence intensity of hydrogel glucose sensors by backscattered frequency modulation. The frequency offset of the backscattered signal highly correlates with the glucose concentration. In conclusion, the proposed mm sized wireless sensor system with fluorescent glucose sensors shows the feasible application for a comfortable, biocompatible, and stable method for the CGM. We further envision that our backscattered frequency-modulation approach can be applied to other fluorescence-based sensors in general, thereby facilitating paradigm shift to in vitro continuous vital-signal monitoring.

## Figures and Tables

**Figure 1 sensors-21-08464-f001:**
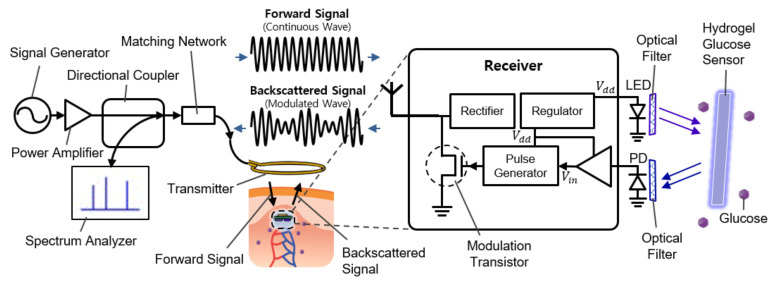
Schematic illustration of the implantable continuous glucose monitoring system, including implantable hydrogel sensor, implantable frequency modulation module, and wearable transmitter.

**Figure 2 sensors-21-08464-f002:**
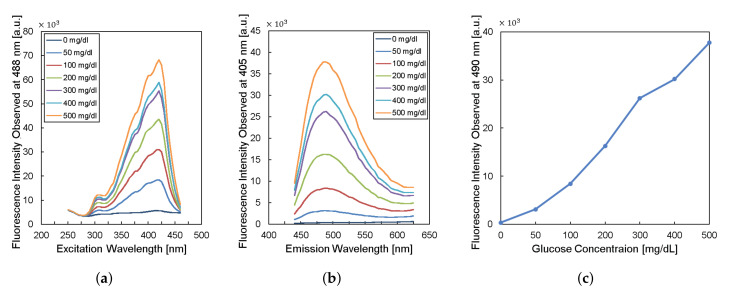
Characteristics of fluorescent monomer response to the glucose levels. (**a**) The excitation profile and (**b**) the emission profile of the fluorescence sensor. (**c**) The fluorescence intensity observed at the wavelength of 490 nm versus glucose concentration.

**Figure 3 sensors-21-08464-f003:**
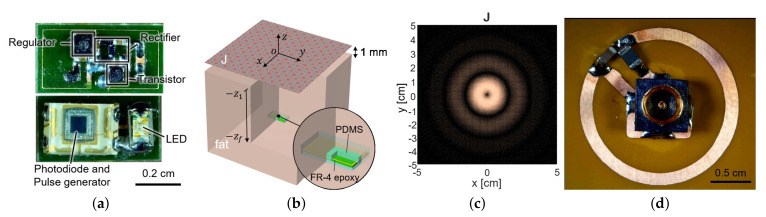
(**a**) The fabricated sensor with the coil and the circuit components. It contains a loop antenna, matching network, rectifier, storage capacitor, and power transistor for shifting impedance (top) and photodiode and LED (bottom). (**b**) The planar model consists of a source surface current density J and a receiver coil placed within a tissue. The surface current source and the air–tissue interfaces are at the plane of z=0 and $z=-z_{1}$, respectively. The receiver is embedded at the depth of $\left(z=-z_{f}\right)$. (**c**) The optimal current distribution at 1.5 GHz. (**d**) A coil structure for the transmitter that realizes the optimal current distribution.

**Figure 4 sensors-21-08464-f004:**
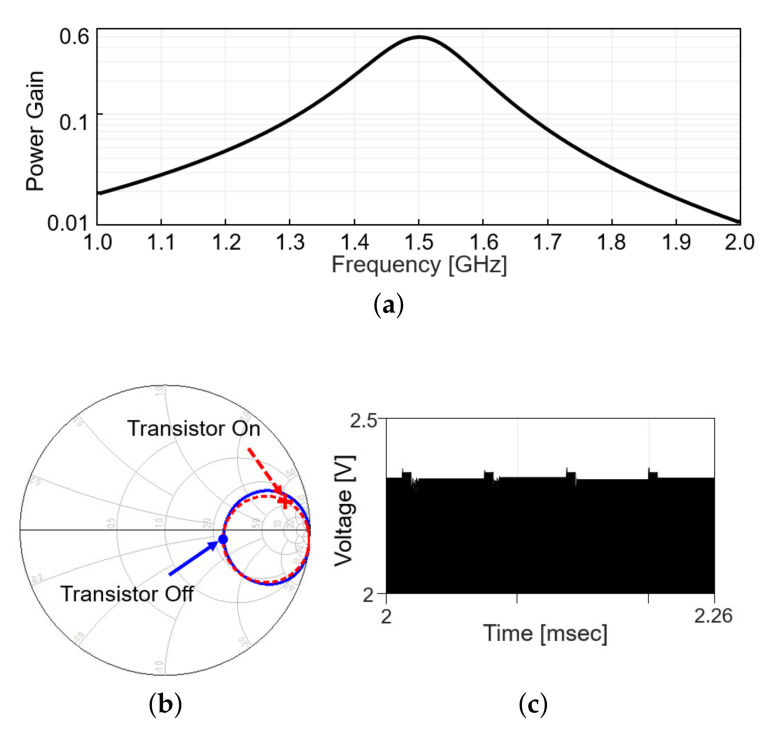
Wireless link analysis in simulation. (**a**) Power gain plot is obtained by simulation that includes a s2p file modeling the coupling between Tx and Rx coil and matching networks. (**b**) Smith chart for $S_{11}$ at the transmitter with the frequency range from 1 to 2 GHz. (**c**) Backscattered signal in time domain.

**Figure 5 sensors-21-08464-f005:**
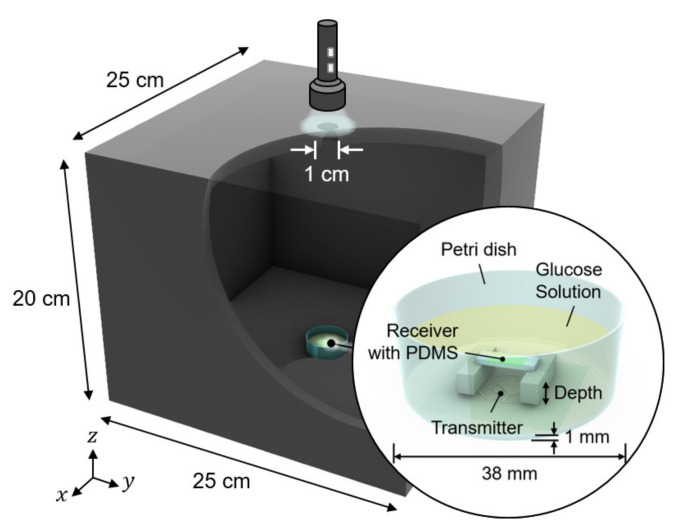
Measurement setup for wireless link analysis. The measurement was conducted in a dark chamber to avoid the interference from ambient light.

**Figure 6 sensors-21-08464-f006:**
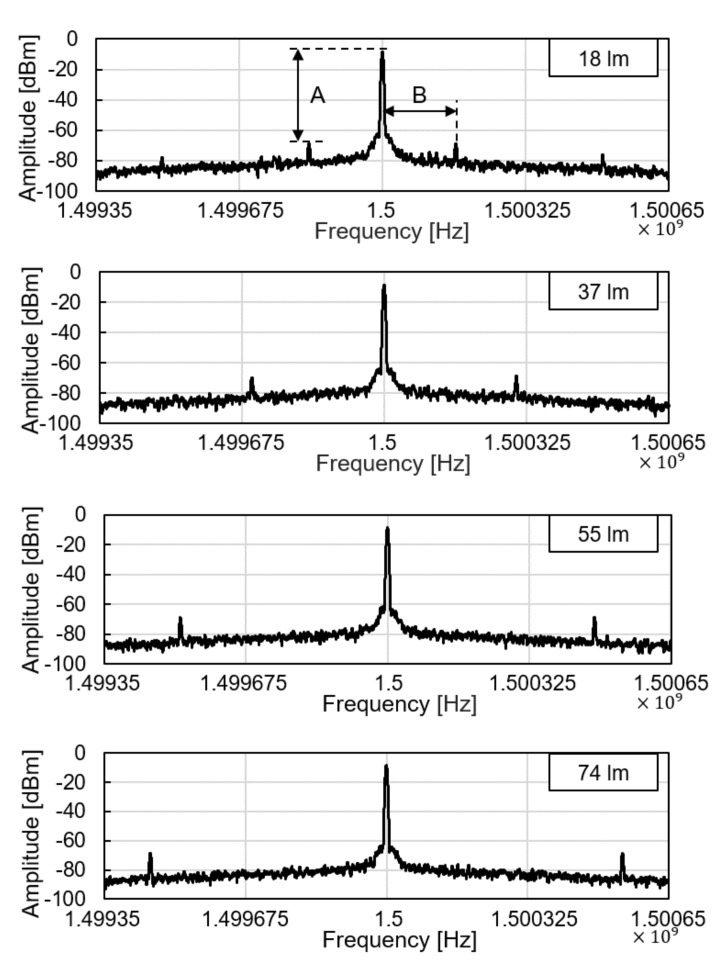
Backscattered signal measured by a spectrum analyzer with varying light intensity.

**Figure 7 sensors-21-08464-f007:**
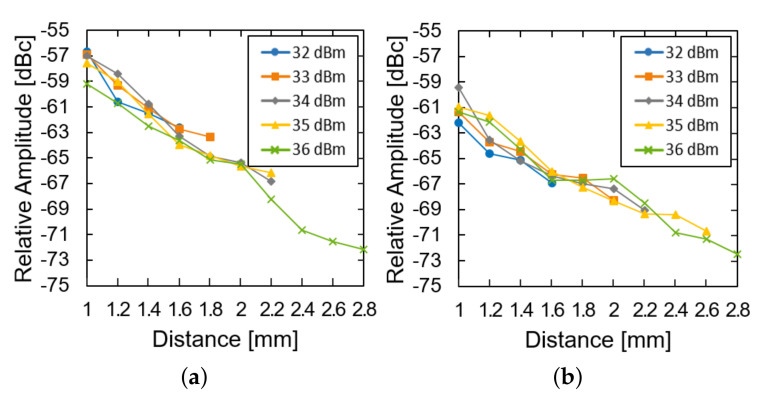
Measured results of the subtraction of the modulated signal from the carrier signal (**a**) in the air and (**b**) in the glucose solution according to the distance and transfer power.

**Figure 8 sensors-21-08464-f008:**
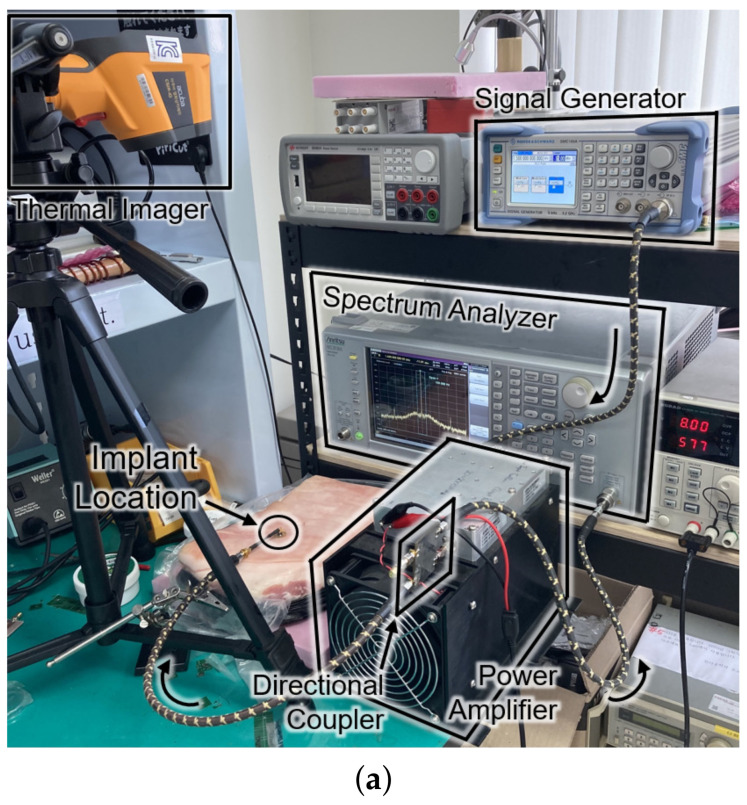
(**a**) Photograph of the safety test setup within a piece of pork. (**b**) Top view of the pork with an infrared thermal image before (top) and after (bottom) powering for a duration of 20 s. (**c**) Cross section of the SAR simulation according to the input power.

**Figure 9 sensors-21-08464-f009:**
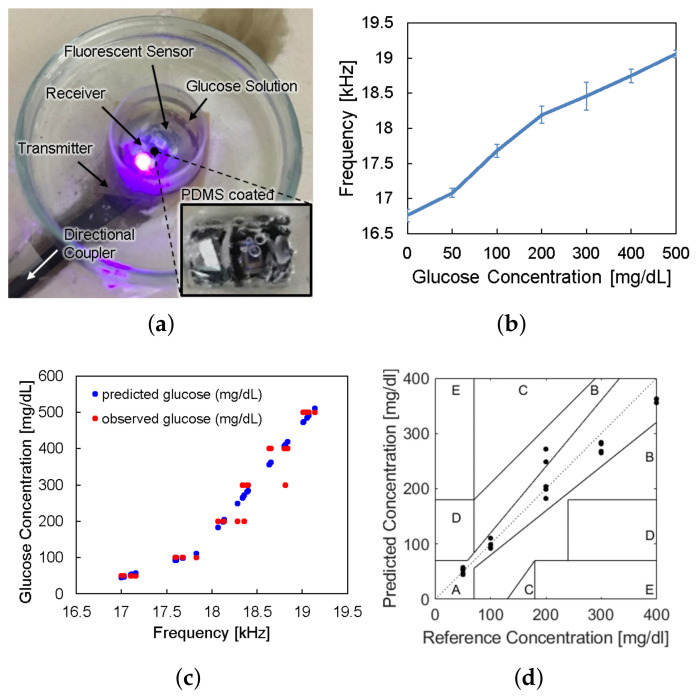
(**a**) Photograph of in vitro glucose-response test setup. (**b**) Measured results of backscattered frequency versus glucose concentration. (**c**) Expectation of glucose level response by backscattered frequency. (**d**) Clarke’s error grid analysis of the whole system test.

## Data Availability

Not applicable.

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
