# Peer review of "Batteryless, Miniaturized Implantable Glucose Sensor Using a Fluorescent Hydrogel"

_sensors, 2021, doi:10.3390/s21248464_

Round 1

Reviewer 1 Report

Review report for sensors-1478433 entitled “Batteryless, Miniaturized Implantable Glucose Sensor Using a Fluorescent Hydrogel”

The authors of the manuscript aimed to realize minimally invasive implantable glucose monitor. In the manuscript, mm-scale fluorescence based glucose chemical sensor was developed and tested in-vitro. Developed devices and behind technology of them is clearly explained. I feel sophistication of research design from the manuscript. However, in the point of view from the sensors side witch I expertized, I have some question and comments would like to ask to the authors.

Q1. At line 23, the authors said that “…sensors suffers low accuracy and low correlation….”. Please quantify the requirement for glucose sensors on accuracy and etc. that the author believes the device should have.

Q2. Related to line 34, Is a design goal of 3 months or longer usable time appropriate for an implantable device that imposes a burden on the patient for replacement?

Q3. Representation of Fig.2 can be improved. I think (a) shows excitation spectrum, not absorbance spectrum. If so, authors should show fixed observation fluorescence wavelength. Also, (b) should have fixed excitation wavelength. 

To summarize the above, 

Fig,2a y axis: “Fluorescence intensity observed at XXX nm (a.u.)”.

Fig.2a x axis; “Excitation wavelength (nm)”

Fig,2b y axis: “Fluorescence intensity excited at XXX nm (a.u.)”.

Fig.2b x axis; “Emission wavelength (nm)” 

Q4. The frequency shift of the backscattered signal observed in the experiment shown in Fig. 6 means that the influence of ambient light is significant. When the sensor is implanted, will disturbing light enter the sensor and affect the determination of glucose concentration?

Q5. At Line 249, The authors tested with “physiological blood-glucose concentration”, however, glucose concentration in interstitial fluid should be lower than in blood. I believe that authors have to tested with glucose concentrations at interstitial range.

Q6. In Figure 9, the “PDMS coated” part is blurred and cannot be seen clearly, so it would be better to replace the image if possible.

Reviewer 2 Report

In this manuscript, the implantable device for monitor of glucose has been developed. The amount of glucose was detected via the fluorescence, which is increased at the higher concentration due to suppression of photo-induced electron transfer. The data was transferred via not a wire but a loop antenna. The authors have focused on the wireless transferring of data from the implant.

The proposal and feasibility study are interesting and thus publishable after the revision and/or reply to the comments as follows. 

In Fig. 9b, the backscattered frequency shows a sigmoidal curve, while the fluorescence may be linearly increased at higher concentration of glucose (see Fig. 2). Can the signal intensity detected by the photodiode in the device, which may correspond to Vin in Fig. 1, be derived from the backscattered frequency in Fig. 9b for comparison with the fluorescence intensities in Fig. 2b? 

Although the references for section 2.1 are cited, the information (name of the reagents and so on) of the fluorescent monomer in the present study should be written in detail. 

In Line 20, "and" is duplicated. 

Round 2

Reviewer 1 Report

I grateful to the authors for addressing all the questions and comments. I believe revised manuscript is ready to publish.

Author Response

Thank you for recommending the acceptance. We revised the manuscript a little according to reviewer 2's comment.

Reviewer 2 Report

This manuscript has been properly revised. The new experimental result with the longpass filter shows the good correlation compared with the previous result. But, the correlation between the fluorescence intensities and glucose concentrations, which displayed in the author response, can be shown in this manuscript as Fig. 2c.  
